# Emergent spatial structure in the gut microbiota is driven by bacterial growth and gut contractions

**Giorgia Greter[1]\*, Sebastian Hummel[1,2], Daria Künzli[1], Naomi Dünki[1], Niina Ruoho[1], Patricia Burkhardt[1], Suwannee Ganguillet[1], Milad Radiom[1], Claudia Moresi[1], Leanid Laganenka[3], Wolf-Dietrich Hardt[3], Steffen Geisel[4], Julien Bauland[4], Sebastian Jordi[5,6], Benjamin Misselwitz[5,6,7], Bahtiyar Yilmaz[5,6], Jonasz Słomka[8], Eleonora Secchi[8], Roman Stocker[8], Emma Slack[1,9,10,11], Markus Arnoldini[1]\***

**1** Department of Health Sciences and Technology, ETH Zurich, Zurich, Switzerland, **2** Institute of Mathematics, BOKU University, Vienna, Austria, **3** Department of Biology, ETH Zurich, Zurich, Switzerland, **4** Department of Materials, ETH Zurich, Zurich, Switzerland, **5** Department of Visceral Surgery and Medicine, Bern University Hospital, University of Bern, Bern, Switzerland, **6** Maurice Müller Laboratories, Department for Biomedical Research, University of Bern, Bern, Switzerland, **7** Department of Medicine II, Hospital of the LMU Munich, Munich, Germany, **8** Institute for Environmental Engineering, Department of Civil, Environmental, and Geomatic Engineering, ETH Zurich, Zurich, Switzerland, **9** Sir William Dunn School of Pathology, University of Oxford, Oxford, United Kingdom, **10** Basel Research Center for Child Health, Basel, Switzerland, **11** Botnar Institute for Immune Engineering, Basel, Switzerland

\* giorgia.greter@hest.ethz.ch (GG); markus.arnoldini@hest.ethz.ch (MA)

## Abstract

Spatial structure can functionally determine ecological interactions and evolution of microbial communities. The gut microbiota is known to be spatially structured longitudinally along the gastrointestinal tract, but micro-scale structure in the gut lumen has not been extensively explored. Here, we show that bacteria cluster within species in the cecum of gnotobiotic mice. We find that clustering is not driven by active swimming, antibody-mediated aggregation, or factors exclusive to the host, but likely due to bacterial growth in the matrix of gut content. In samples from mice and humans, we show that upper large-intestinal content behaves as a nonNewtonian fluid that changes its viscoelastic properties under the force of gut contractions. We argue that microbial growth in the gel-like structure of cecum content can lead to micro-scale bacterial clustering, which is periodically disrupted by peristalsis-driven shear thinning and clearance. Our study shows mechanistically how spatial structure in the gut emerges through the interplay between microbial and host physiology and highlights the possibility of host control over gut microbiota distribution through gut contractions.

**Data availability statement:** All data relevant for generating figures is available in a data repository under https://doi.org/10.5281/zeno-do.19422141. This data repository includes annotated scripts for analyzing data and generating the plots which underlie the figures in this publication, as well as a detailed instructions. Data was analyzed and plotted using R V4.3.2 and Python 3.11.4.

**Funding:** This work was supported by the Swiss National Science Foundation (project grants 1851228 to ESI and 320030_197815 to BM, Spark award CRSK-3_220620 to MA, Ambizione grant PZ00P2_202188 to JS, PRIMA grant 179834 to ESe, Starting Grant TMSGI3_211300 to BY) and Innosuisse (120.452 IP-LS to MA). It was also supported as part of NCCR Microbiomes, a National Centre of Competence in Research, funded by the Swiss National Science Foundation (180575 to ESI). The funders played no role in study design, data collection and analysis, decision to publish, or preparation of the manuscript.

**Competing interests:** The authors have declared that no competing interests exist.

**Abbreviations:** FISH, fluorescent in-situ hybridization; MFI, mean fluorescent intensity; sIgA, secretory Immunoglobulin A; SPF, specific pathogen-free.

## Introduction

The gastrointestinal microbiota plays important roles in maintaining host health and metabolic homeostasis [1,2]. Microbiota density and composition vary along the gastrointestinal tract, due to strong changes in environmental factors such as pH, flow rates, and oxygen levels [3–6]. In the dense bacterial community in the large intestine, bacteria are not well mixed, even at length scales where such global parameters are constant [7–11]. Given that both positive [12] and negative [13] interactions between bacteria take place in their immediate proximity (interaction ranges of around 10 µm [12]), the direct neighborhood of bacteria is important in determining how they live. Understanding the extent and origins of spatial structure in the gut microbiota is thus crucial for mechanistically understanding ecological and evolutionary processes in this important microbial community.

There are different possible mechanisms for how spatial structure in the large intestine could emerge. Anatomical features, such as colonic crypts, the mucus layer in the distal colon, and oxygen gradients close to the epithelium can create spatially distinct habitats [10,14–17], but the bulk of bacterial growth happens in the gut lumen of the proximal large intestine without such pre-defined structural features [6,9,18]. Nevertheless, micro-scale spatial structure can emerge in the gut lumen, and possible mechanisms include small-scale differences in physical properties (e.g., density, viscosity, and adherence to food particles) of luminal content [19–21], cross-linking of bacteria by antibodies [22], and flagella-driven chemotactic movement of bacteria [23,24]. If the environment prevents dispersal and nutrients are available, bacterial cell division can generate patches of clonal cells [25]. The size of such microcolonies will depend on the niche size, as bacterial growth depends on the ecological space and resources available in a niche, as well as the stability of that niche [26–28].

All the processes that can lead to spatial structure in the gut lumen take place in, and interact with, the complex matrix of gut content. Gut contractions, which play an important role in maintaining microbial abundance and composition [6,29,30], exert forces on the gut lumen when propelling and mixing luminal content. Intestinal content is shear thinning (i.e., it displays decreasing viscosity with increasing shear rate), and changes its properties from more solid-like to more liquid-like when a contractile force is applied (i.e., it undergoes a yielding transition) [31–34]. This may render bacterial swimming, nutrient distribution, and more generally the possibility of forming spatial structure in the lumen dependent on gut contractions: particulates and cells can move more easily when gut content behaves liquid-like, while they are kept in place when it behaves solid-like.

Here, we quantify the influence of different possible mechanisms governing bacterial spatial dynamics in the mammalian gut. While chemotaxis and antibody binding play detectable roles, we conclude that bacterial growth is a major factor in determining bacterial clustering in the gut. In the matrix of gut content, with rheological properties that restrict movement in the absence of contractile force, bacterial cell division can result in microcolony formation. The extent of this microcolony formation is limited by gut contractions [7–11]. By experimentally modulating flow and mixing in the gut lumen of mice, and by quantifying rheological parameters of gut content from

mice and humans, we reveal how the host can change the biophysical properties of the gut lumen via gut contractions, potentially giving it control over the extent of microcolony formation and thus ecological interactions in the gut microbiota.

## Results

### Organ-scale spatiotemporal dynamics of mixing in the murine cecum

To experimentally assess the strength and dynamics of mixing in the mouse cecum, the part of the large intestine with most bacterial activity, we have analyzed the distribution of bacteria in two parts of the organ: at the cecum tip (the part of the cecum furthest away from the ileum and colon) and the cecum base (the part of the cecum close to where ileum and colon connect to it) (Fig 1A). To allow full visualization of the distribution of different microbiota members, we used a gnotobiotic mouse model containing three bacterial species (three-member microbiota, 3MM [35]), *Bacteroides thetaiotaomicron* (*B. theta*)*, Eubacterium rectale* (*E. rectale*)*,* and *Escherichia coli* (*E. coli*) (Figs 1B and S1A). Bacteria in fixed cryosections of cecum tissue and content were visualized using fluorescent in-situ hybridization (FISH) and confocal microscopy, and center coordinates for every bacterium were determined using a custom image analysis method (S1B Fig, Materials and methods).

We found that the composition of the microbiota differed in the two locations within the same organ. We found significantly more *B. theta* and *E. coli* at the cecum tip, and more *E. rectale* mostly at the cecum base, with large variation between analyzed images (Fig 1C). This indicates that the gut content is not uniformly mixed within a single gnotobiotic cecum. To evaluate whether this intra-cecum heterogeneity can be generalized beyond our gnotobiotic model, cecum tip and base content were harvested from OligoMM$^{12}$ mice containing 12 bacterial species [36], and specific pathogen-free (SPF) mice harboring an even more complex microbiota. We determined bacterial community composition in the different parts of the organs using 16S sequencing (S2A, S2B Fig). Differences in community composition were analyzed using Jensen-Shannon distances at the family-level (S2C, S2D Fig). While variation in microbiota composition was generally large, family-level Jensen-Shannon distances were significantly larger between parts of the cecum than between cecum tips of the cecum OligoMM$^{12}$ mice (S2E Fig), and in SPF mice they were significantly larger between bases than between tips of different ceca (S2F Fig). These findings indicate that, in mice with different microbiota complexities, microbiota composition can vary among different parts of the cecum.

Given these differences in bacterial composition at different locations of the cecum, we sought to evaluate how material is transported through the cecum. We orally administered bacteria-sized (1 μm diameter) fluorescent beads to SPF mice at three different time points 40 min apart (Fig 1D). To distinguish beads administered at different times, beads of different colors were used at every time point, and every mouse received one bolus of beads at time 0 min, one at 40 min, and one at 80 min (Fig 1D). Five hours after the first treatment, the distribution of beads was evaluated (Figs 1E and S3). In three out of five mice, the bead abundance in the base dropped monotonously from 220 to 300 min, whereas in the other two it increased initially and dropped at the last time point (Fig 1E, red bars). The opposite is true for the tip, where relative bead abundance increased monotonously in three out of five animals, while it dropped at 260 min and then increased at 300 min in two (Fig 1E, blue bars). Even though bead distribution varied considerably between mice, there are some key insights that can be gained from this data. Our results show a spatiotemporal pattern for particle distribution in the mouse cecum: after entering the cecum lumen at the base, particles moved to the cecum tip on a time scale of hours. The data further suggests that particles move along the epithelium, and only later disperse to the center of the cecum lumen, as relative counts at the tip epithelium increase earlier than counts in the tip center in three out of five mice, and no beads are observed in the tip center in the other two (light versus dark blue bars in Fig 1E).

### Bacteria show more micro-scale clustering than micrometer-sized beads in the lumen of the murine cecum

We next investigated the spatial structure of the microbiota in the cecum lumen at a micrometer-scale. We determined whether bacterial spatial distribution is random, clustered or regular, using the inhomogeneous H(r) function, a derivative of Ripley's K-function [37]. In short, this method is based on counting the number of cells found in circles of increasing

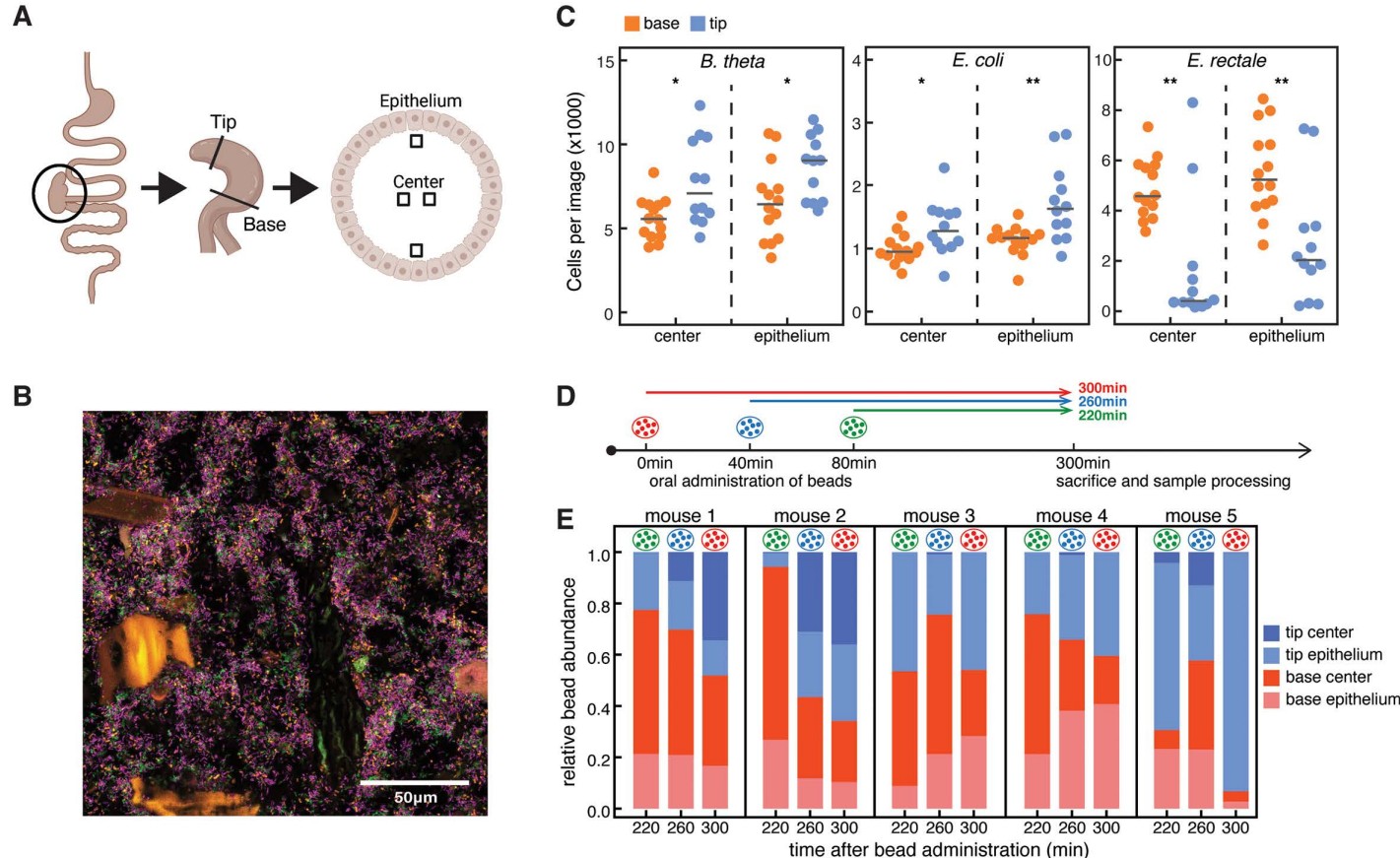

**Fig 1. Bacteria are not homogenously mixed throughout the murine cecum. (A)** Schematic illustrating the experimental procedure for capturing images. The entire cecum is fixed and frozen, and cryosections are produced for tip and base. Four images were taken for each section: two at the center and two at the epithelium. **(B)** Representative microscopy image of the 3MM gnotobiotic model taken at the center base. Green cells are *B. theta,* orange cells are *E. coli,* purple cells are *E. rectale.* Autofluorescent large food particles were removed in the process of analyzing images. **(C)** Number of cells per image in the 3MM mouse, comparing each location between the base and the tip. There is a significant difference between base and tip for all imaged locations and all 3MM species (*t* test, * = $p < 0.05$, ** = $p < 0.01$). Black lines indicate mean. **(D)** Schematic setup for the experiment yielding the data shown in **(E)**. Mice were administered beads that were fluorescently labeled with three different fluorophors. At 0 min red beads, at 40 min blue beads, and at 80 min green beads were orally gavaged into each animal in the experiment. Mice were subsequently sacrificed at 300 min, and bead distribution was analyzed using confocal microscopy. (E) Relative bead abundance in different sections of the cecum 220 min (green beads), 260 min (blue beads), and 300 min (red beads) after gavage of beads, for 5 mice, following the experimental setup shown in (D). Red beads were gavaged 300 min, blue beads 260 min, and blue beads 220 min before the mice were sacrificed. The image in (A) was created using BioRender (https://bioren-der.com/x3jgf5c). The data underlying this Fig can be found in the data repository accompanying this paper under the DOI: https://doi.org/10.5281/zenodo.19422141.

radius r around a focal cell and comparing this to the case in which the distribution is random (i.e., follows a Poisson process; see Materials and methods for details). In our Figures, H = 0 indicates random distribution, H > 0 indicates clustering, and H < 0 indicates regularity (Fig 2A).

We found that all three species of the 3MM were clustered (Fig 2B, showing pooled data for all measured locations), regardless of their location in the cecum (S4 Fig). In addition to a qualitative analysis of clustering, the results of the H function can be employed to estimate the radius of a typical cluster [37]. Specifically, the value maximizing the H function roughly corresponds to the diameter (or twice the radius) of a typical cluster. Thus, our analysis suggests clusters with radii of around 3–4 µm (Fig 1B).

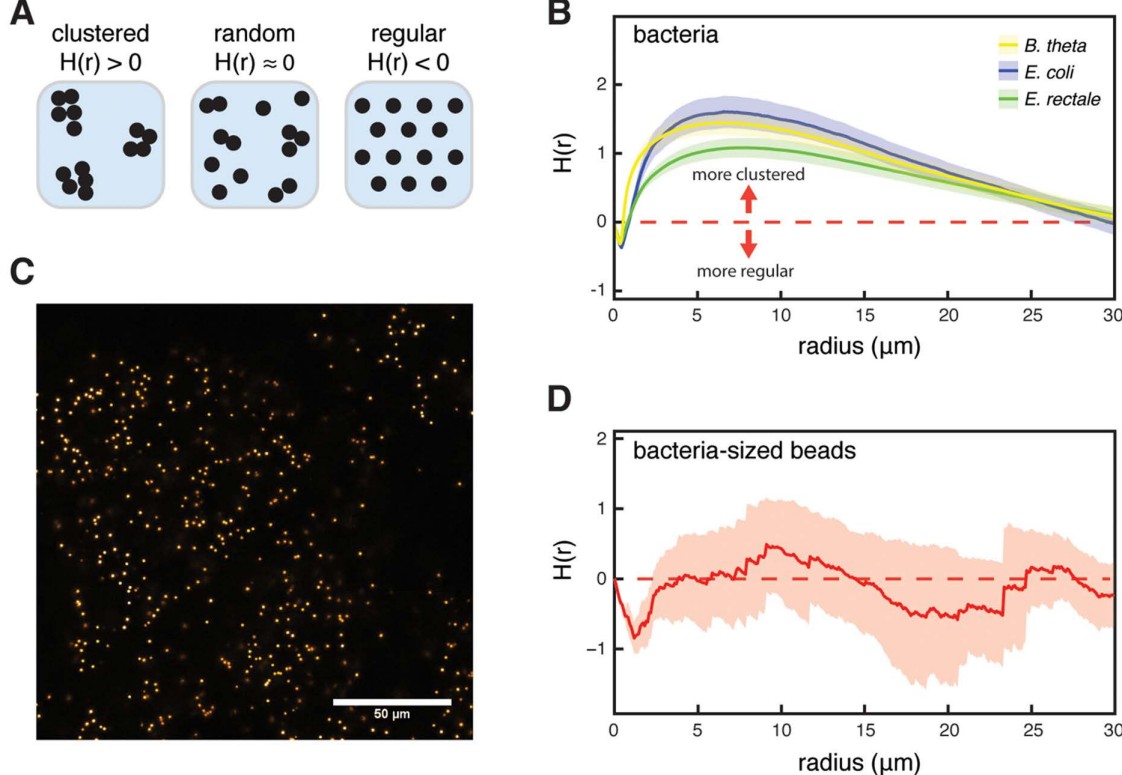

**Fig 2. Bacteria display clustering spatial point patterns in the murine cecum. (A)** Schematic illustration for three possible spatial point patterns: clustered, complete spatial randomness, and regular. **(B)** Results for all three 3MM species when cell distribution was analyzed using the inhomogenous H function. Shaded region indicates 95% CI ($n=6$). Red dashed line indicates the theoretical value for complete spatial randomness. All bacterial species show significant clustering ($p<0.01$, Studentized permutation test with Bonferroni correction). Data is shown for all images taken at the cecum base. **(C)** Confocal microscopy image of fluorescent beads in the 3MM base center. Beads are 1 µm in diameter. **(D)** Results for fluorescent bead distribution in the 3MM mouse cecum, analyzed using the inhomogeneous H function ($n=6$). Beads are 1 µm in diameter, data shown for cecum base. Shaded region indicates 95% CI, dashed red line is the theoretical value for complete spatial randomness. Bacteria-sized beads do not show significant clustering in the 3MM mouse cecum ($p=0.736$, Studentized permutation test with Bonferroni correction). The data underlying this Figure can be found in the data repository accompanying this paper under the DOI: https://doi.org/10.5281/zenodo.19422141.

Having found that bacteria cluster in the cecum of 3MM mice, we set out to test possible causes of this phenomenon. To test whether abiotic factors, such as micro-scale differences in viscosity or transport dynamics, lead to clustering independent of bacterial activity, we fed $10^9$ 1-µm-diameter fluorescent beads to 3MM mice. These beads approximately mimic the size of single bacteria (Fig 2C). We analyzed bead clustering in the cecum 4.5 h after treatment. In the cecum base, we found no significant clustering of beads (Fig 2D; data for cecum base, no beads were found in the cecum tip, likely due to the larger size of the cecum in 3MM mice; when analyzing center and epithelium data separately, beads do exhibit significant clustering at the epithelium, but this clustering is weaker than what we observe for *B. theta* and *E. coli*, S4 Fig). This indicates that bacterium-sized, abiotic particles are well mixed on a microscopic scale, at least at the cecum base, and therefore that bacterial activity is necessary for the clustering of cells that we observe in the cecum.

## Antibodies and chemotaxis are not responsible for bacterial cluster formation

As a next step, we tested the role of several biological functions that could play a role in bacterial clustering.

First, we investigated the role of secreted immunoglobulin A antibodies (sIgA) in cluster formation. High avidity sIgA targeting bacterial surface antigens has been shown to enchain bacteria upon division, which can result in bacterial clustering that depends on bacterial growth rates [22,38]. We found that ex-germ-free mice colonized for 4 weeks with the 3MM strains produced specific sIgA against *E. coli* and *B. theta*, but not *E. rectale* (Fig 3A, red dots). Interestingly, mice which were bred with the 3MM microbiota and were thus colonized from birth had no or very weak specific sIgA production (Fig 3A, blue dots), suggesting a role of early life exposure in altering the interaction between gut microbes and the adaptive immune system. Microscopic bacterial counts were similar in 3MM-bred mice and ex-germ-free mice, indicating that sIgA did not reduce bacterial population sizes, consistent with the concept that sIgA only drives elimination of bacteria in the context of niche competition [39,40] (S5A Fig). When comparing *E. coli* clustering in 3MM-bred mice with ex-germ-free mice colonized for 4 weeks, analysis of the H functions of the images using a Studentized permutation test [41] revealed a significant difference between 3MM-bred and ex-germ-free mice (Fig 3B, 3C), indicating an increase in cluster size and density in the ex-germ-free mice. As we expect bacterial replication and clearance rates in both groups of mice in this experiment to be similar, the observed increase in aggregate size is quantitatively small and likely due to the phenomenon that sIgA-crosslinked clusters are harder to break apart during peristalsis. These dense clusters were visible on microscopy images (Fig 3C). However, as clustering is also observed in the absence of detectable IgA responses, we conclude that clustering can occur independently of sIgA.

A second factor that could contribute to bacterial clustering is chemotactic motility [42]: bacteria might actively congregate following chemotactic cues. To test whether this bacterial trait affects clustering, we monocolonized germ-free mice with one of three *mCherry* fluorescent *E. coli* strains: one that lacks the ability to perform chemotaxis (Z1331::*cheY*), one that lacks functional flagella (Z1331::*flhD*), and the wild type Z1331 as a control. All three strains colonized germ-free mice (Fig 3D), and we evaluated the extent of bacterial clustering in the cecum in these three cases. Statistical analysis showed no significant difference in clustering between wild type and chemotaxis or flagella deficient *E. coli* cells at the center of the cecum base (Fig 3E, 3F). At the epithelium of the cecum base, clustering in images with Δ*flhD* mutant bacteria was significantly different than in wild type, suggesting that *E. coli* likely uses flagella in the gut. However, the H(r) function shows stronger clustering in Δ*flhD* than in wild type *E. coli* (S5B Fig), indicating that motility is not the origin of clustering, but could rather contribute to cluster dispersal.

## Clustering increases with niche size in the mouse cecum

The next variable that we analyzed as a possible cause of bacterial clustering was the total population size of a given bacterial species in the cecum, the niche size. Given constant loss of bacteria from the GI tract, the relative population size of a given bacterium is a direct readout for the amount of biomass this bacterial population has to produce to maintain a steady state [43]. While this does not lead to more divisions per cell under steady state conditions, denser populations increase the likelihood of starting clusters closer together. In addition, larger clusters might lead a higher probability that clustered cells are lost from the cecum together, leaving a larger part of the niche open for the remaining cells to fill by growth. Both these phenomena should lead to weaker clustering for bacteria occupying smaller niches. To test this hypothesis, we experimentally decreased the *B. theta* niche relative to 3MM mice by colonizing OligoMM$^{12}$ mice with *B. theta.* Since the OligoMM$^{12}$ microbiota contains 12 species (but not *B. theta*), it presumably contains fewer available niches for *B. theta* to occupy than the 3MM microbiota. Accordingly, *B. theta* counts were >2-fold lower in OligoMM$^{12}$ mice compared to 3MM mice in all studied sections of the cecum, presumably due to a smaller available metabolic niche when competing with the OligoMM$^{12}$ microbiota (Fig 3G). While *B. theta* remained clustered, the extent of clustering was significantly reduced in OligoMM$^{12}$ mice as compared to 3MM mice (Fig 3H, 3I). The results indicate that niche size, and consequently bacterial growth, significantly influence the formation of bacterial clusters in the mouse cecum. This finding suggests that these clusters may represent clonal microcolonies that develop in cecum content that is transiently unmixed between gut contractions.

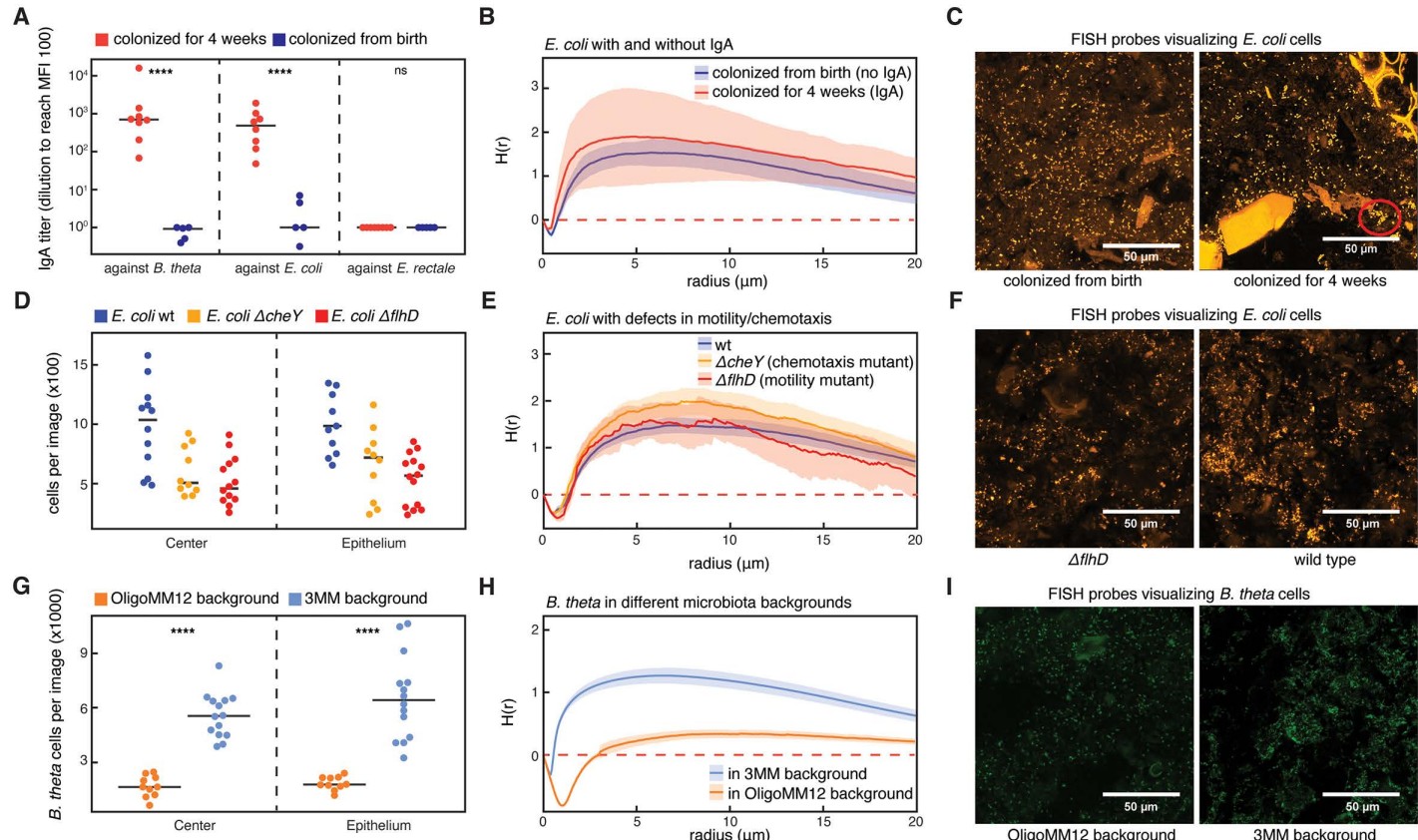

**Fig 3. The influence of biotic factors on bacterial cluster formation.** (A) 3MM-specific IgA titers in mice bred with the 3MM microbiota ($n = 5$) and ex-germ-free mice colonized with the 3MM strains for 4 weeks ($n = 8$). Specific IgA is measured by testing specific binding in small-intestinal lavages against bacterial cultures using flow cytometry. Titers are then calculated by estimating the number of dilutions required to reach a median fluorescence intensity of 100 ($t$ test, **** = $p < 0.0001$). (B) Results for spatial distribution of $E.\ coli$ cells in mice bred with the 3MM microbiota (blue), and ex-germ-free mice colonized with the 3MM microbiota for 4 weeks (red), using the inhomogeneous H function. Both groups show significant clustering ($p > 0.01$), but their respective spatial distributions are not significantly different ($p = 0.513$). Data for cecum tip, center. (C) Representative confocal microscopy images with FISH-stained $E.\ coli$ cells in fixed cecum content, from mice colonized from birth (left) and mice colonized for 4 weeks (right). A tight cluster that is presumably formed by sIgA-cross-linking is marked by a red circle in the image on the right. (D) Counts of wild type $E.\ coli$ (wt, blue), chemotaxis mutant ($\Delta cheY$, yellow), and motility mutant ($\Delta flhD$, red) in ex-germ-free mice on microscopy images. Counts are measured at the cecum base. (E) Results for spatial distribution of wt (blue), $\Delta cheY$ mutant (yellow), and $\Delta flhD$ mutant $E.\ coli$ cells in the cecum ex-germ-free mice, using the inhomogeneous H function ($n = 12$, 4 in each group, cecum base center data shown). All groups show significant clustering ($p < 0.01$), and there is no significant difference in clustering between groups ($p > 0.05$). (F) Representative confocal microscopy images with FISH-stained $E.\ coli$ cells in fixed cecum content, from mice that were monocolonized with the $\Delta flhD$ $E.\ coli$ mutant lacking flagellar activity, and wild type $E.\ coli$. (G) Counts of $B.\ theta$ cells in the context of the OligoMM$^{12}$ (orange, $n = 5$ mice) and 3MM microbiota (blue, $n = 6$ mice). Data for cecum base ($t$ test, **** = $p < 0.0001$). (H) Results for spatial distribution of $B.\ theta$ cells in OligoMM$^{12}$ (orange) and 3MM (blue) microbiota background, using the inhomogeneous H function. $B.\ theta$ clustering is significantly reduced in OligoMM$^{12}$ as compared to 3MM mice ($p < 0.01$). (I) Representative confocal microscopy images with FISH-stained B. theta cells in fixed cecum content from mice colonized with $B.\ theta$ and the OligoMM12 microbiota (left) and the 3MM microbiota (right). Shaded regions in (B), (E), and (H) indicate 95% CI, dashed red lines are the theoretical value for complete spatial randomness. Bonferroni-corrected Studentized permutation tests were used for statistical comparisons between H(r) functions. The data underlying this Figure can be found in the data repository accompanying this paper under the DOI: https://doi.org/10.5281/zenodo.19422141.

## Rheological properties of cecum content could allow microcolony growth between mixing events

To investigate how the physical properties of cecum content and the force exerted by gut contractions might influence microcolony formation, we analyzed the viscoelastic properties of fresh cecum content using rheometry. Our observation that beads traverse the cecum close to the epithelium (Fig 1D) could be attributed to a shear thinning behavior of the

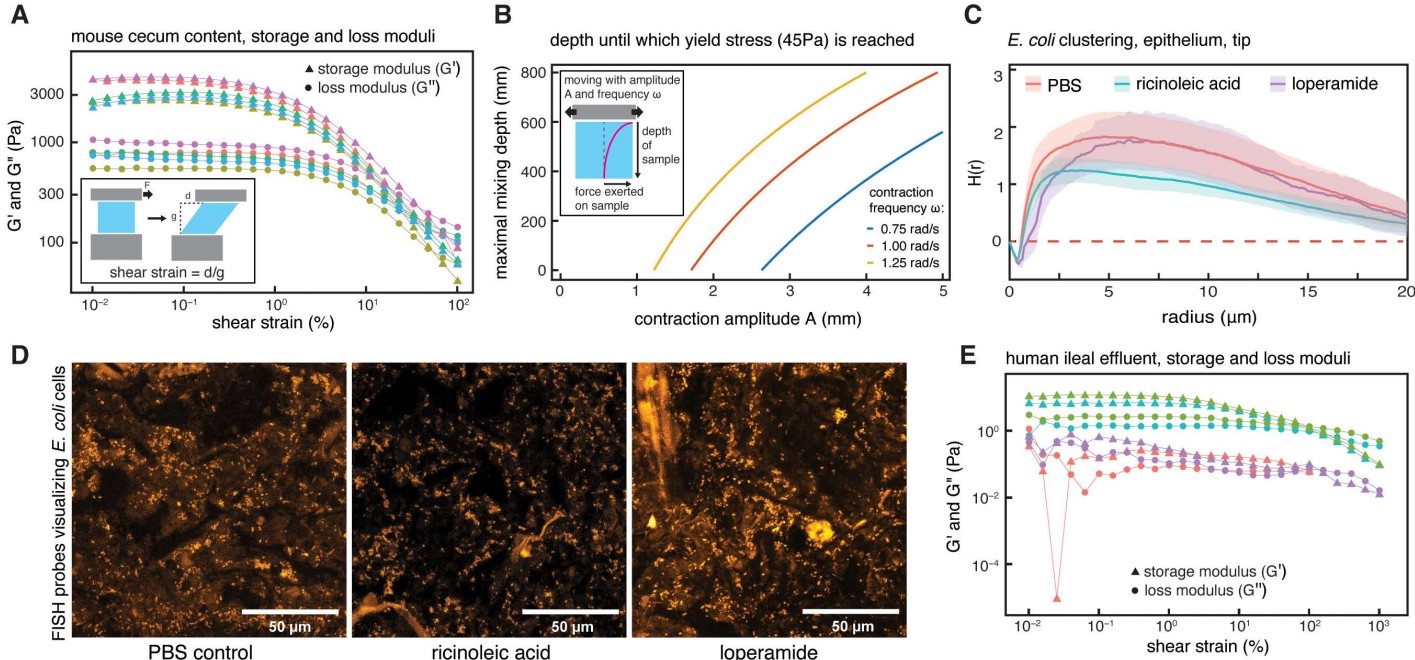

cecum content (i.e., decreasing viscosity when force is applied), facilitating movement along the epithelial walls, close to the smooth muscle layer and mucus-secreting goblet cells. To investigate this, we measured the viscoelastic properties of 3MM mouse cecum content. Using a parallel-plate, rotational rheometer, we applied oscillatory shear strains at defined amplitudes (the shear strain is defined as the horizontal displacement of the tested material, d, relative to the gap between the plates in the rheometer, g, Fig 4A, inset) and a constant oscillation frequency of 1 rad/s. The resulting force exerted at every given shear strain (the shear force) is dependent on the material's viscoelastic properties. We found that shear stress increases with increasing strain amplitude up to a certain threshold, the yield limit (S6A Fig). At shear strains higher than the yield limit, shear strain has less effect on shear stress. These findings indicate that cecum content behaves solid-like at low shear stress up to the yield limit, and as a viscoelastic fluid at higher shear stress. In rheology, this behavior is typically illustrated using the storage and loss moduli (G' and G'', respectively, Fig 4A). Conceptually, G' represents the elastic or stored energy in a material when it is deformed, whereas G'' represents the viscous or dissipated energy during deformation. Where these two quantities intersect, the material transitions from more solid-like (i.e., dominated by G', triangle symbols in Fig 4A) to more liquid-like (dominated by G'', circle symbols in Fig 4A). Different approaches exist to quantitatively estimate this yield limit (S6 Fig) [44]. Using the measurements from oscillatory rheology (Fig 4A),

**Fig 4. Physical properties of cecum content can influence clustering. (A)** Storage (G', triangles) and loss (G'', circles) moduli of mouse cecum content. Colors indicate different samples (i.e., cecum content from different mice, *n* = 5). Inset Figure illustrates the definition of shear strain, as the fraction of displacement, d, divided by the gap between plates in the rheometer, g. This quantity is dimensionless and usually reported in %. **(B)** Model results based on Stokes' second problem, predicting the depth a given cecum contraction can reach with enough force to reach the yield limit of 45 Pa (as estimated from our data, S6 Fig), depending on the amplitude and frequency (colors) of contractions. Inset image depicts the underlying process. **(C)** Clustering of *E. coli* in 3MM mice treated with ricinoleic acid, a drug increasing smooth muscle activity, is significantly reduced compared to control mice (*n* = 5 for both groups, *p* = 0.04). Loperamide treatment for 1.5 h (*n* = 4) did not significantly change clustering of *E. coli* when compared to the control group. Shaded areas indicate 95% confidence intervals, red dashed lines indicate theoretical values for complete spatial randomness. Data for cecum tip, center. Studentized permutation tests were used for statistical comparison between H(r) functions and Bonferroni-corrected for the number of comparisons (two in this case). **(D)** Representative confocal microscopy images with FISH-stained *E. coli* cells in fixed cecum content, from 3MM mice treated with PBS (left), ricinoleic acid (middle), or loperamide (right) for 1.5 h. **(E)** Storage (G', triangles) and loss (G'', circles) moduli of human ileal effluent. Colors indicate different samples (*n* = 4). The data underlying this Figure can be found in the data repository accompanying this paper under the DOI: https://doi.org/10.5281/zenodo.19422141.

we approximated the yield limit for mouse cecum content at a shear stress of 46.4 ± 8.7 Pa (intersection of power-law fits of shear strain versus shear stress above and below the yielding point, S6A Fig). Additional estimates using creep rheology on three independent samples of 3MM mouse cecum content resulted in shear stress values of 40, 20, and 45 Pa (35 ± 13.2 Pa, S6C Fig). As shear forces in cecum content are most likely the result of smooth muscle activity in the cecum wall, this physical property of the cecum content may explain microcolony formation of bacteria: during the time between strong contractions, solid-like behavior creates a matrix that keeps bacteria in place and allows microcolonies to form, while contractions liquify the cecum content and disperse clustered bacteria. This idea is in line with recent findings demonstrating that microbial transport and motility depends on local viscosity [45]. Additionally, the highest shear strain will likely occur directly at the contact zone between intestinal content and the gut wall, where muscle activity in the gut wall will affect luminal content most strongly. We therefore expect the strongest shear thinning effect in this area, which would lead to low viscosity, consistent with the observed dispersal of beads along the epithelium into the cecum tip (Fig 1D).

Given these rheological properties of cecum content, we wanted to understand whether gut wall movements at realistic amplitudes and frequencies can exceed the yield limit and thus allow dispersion of microcolonies. We combined our bulk rheology data (Fig 4A) with a simplified model of gut wall movements, based on Stokes Second Problem, to determine the distance from the gut wall at which wall movements are sufficiently strong for shear stress to reach the content's estimated yield limit of 60 Pa, and thus effectively disperse microcolonies. Stokes Second Problem is a classical problem in fluid dynamics that describes the effect of a horizontally oscillating solid surface on an underlying fluid, depending on the fluid's viscoelastic properties [46]. Using realistic values for the amplitude and frequency of gut wall movements (see Materials and methods), we estimated the relevant distance from the wall at which the yield stress can be reached to be in the centimeter range (Figs 4B and S7). While this approach only allows estimating the order of magnitude of the effect of contractions on gut content, it suggests that changing contraction strength could differentially modulate the consistency of content, and thus mixing, throughout the cecum.

To test experimentally whether manipulation of contractile activity in the cecum can affect microcolony formation, we treated 3MM mice with drugs that increase or decrease smooth muscle activity and analyzed bacterial clustering in cecum content 1.5 h after treatment. To avoid a direct effect on bacterial activity and growth, the drugs were administered by intra-peritoneal injection.

Ricinoleic acid stimulates smooth muscle activity [47], and thus increases contractile activity of the cecum walls. Clustering of *E. coli* and *B. theta* at the epithelium of the tip of the cecum was significantly different and consistently lower in ricinoleic acid treated mice than control mice (Fig 4C, cyan versus red curves, Figs 4D, S8). This data indicates that increased smooth muscle activity of the cecum wall can decrease the extent of bacterial cluster formation.

Loperamide is an opioid that decreases smooth muscle activity and slows gastrointestinal transit [48]. However, treatment with loperamide did not show significant effects on clustering of bacteria in the cecum of 3MM mice 1.5 h after treatment (Fig 4C, purple versus red curves, S8B Fig). We suspected that 1.5 h might not be enough time to allow for sufficient bacterial growth to see differences and performed a second experiment where we assessed bacterial clustering 6 h after treatment with loperamide. Surprisingly, we observed a significant reduction of clustering in the mice treated with loperamide for 6 h (S8B Fig, green versus red curves). This effect was more pronounced in the cecum base than in the tip. While this data indicates that decreased smooth muscle activity might not directly lead to increased clustering of bacteria, other mechanisms, such as backflow due to slower colonic passage or diminished bacterial growth due to decreased food intake during loperamide treatment [49] could dominate the observed effect.

## Rheological properties of human ileal effluent would allow transient clustering

After finding that the rheological properties of mouse cecum content may facilitate the formation of transient, clonal bacterial clumps between gut contractions, we investigated whether the same phenomenon might take place in the human

gut. Analogous to our experiments with mouse cecum content, we measured the viscoelastic properties of ileal effluent of four volunteers with ileostomies and observed qualitatively similar behavior to what we observed in mouse cecum content (Fig 4E). We found a diminishing effect of applied shear strain on shear stress, but at values for shear stress that were substantially lower than what we observed in mice, and with more inter-individual variation. G' is higher than G'' at small values of applied shear strain but decreases faster than G'' with increasing shear strain (Fig 4E), indicating that human ileal effluent, similar to mouse cecum content, transitions from a more solid-like to a more liquid-like behavior when force is applied.

The observed quantitative difference in yield stress between cecum content from mice (46.4 ± 8.7 Pa, S6A Fig) and ileal effluent from humans (0.25 ± 0.25 Pa, S6C Fig) may be attributed to the higher water content of ileal effluent compared to colon content, which has been shown to change when the ileum is disconnected from the colon [50]. Previous rheology measurements of human feces have found yield stresses spanning a wide range, from 20 to 8,000 Pa [34]. We expect the rheological properties of human large intestinal content to change considerably along the colon as a function of water resorption, presumably explaining the difference between what we observe for ileal effluent and what has been measured for feces. It is therefore possible that a similar mechanism to the one described in mice could govern microbial assortment in the human large intestine.

## Discussion

Spatial assortment of gut microbiota members is crucial for ecological interactions and evolutionary processes in the gut. In this study, we quantify the spatiotemporal distribution of bacterial cells in the mouse cecum, both on a macroscopic and a microscopic scale, and identify a likely mechanism for the observed behavior.

A first important finding is the distribution of bacteria throughout a whole mouse cecum. We identified a difference in bacterial counts in different parts of the cecum and observed that the main flow from base to tip might be stronger along the epithelium than in the lumen. Subsequent experiments suggest a critical function for the viscoelastic properties of gut content in this phenomenon. Mixing in the mouse cecum generates shear force, which affects gut content most strongly at the epithelium, close to the force's origin. While there is no structured mucus layer covering the epithelial crypts in the mouse cecum, mucus is secreted into the cecum lumen by goblet cells in the cecal epithelium [51]. Higher mucus concentrations close to the epithelium, together with the fact that boundary movement near the walls leads to maximum shear and velocity [52], could therefore allow for maximum bulk transport of particles in this region, running contrary to the current understanding of mucus as a protective barrier at the epithelium [53]. In addition, oxygen gradients across the walls of the cecum might affect the capacity of different bacteria to grow and survive in this area [17].

In 3MM mice, and in contrast to SPF mice, small particles did not reach the cecum tip. A possible factor explaining this observation is the larger size of the cecum in 3MM mice, which could result in a more stretched smooth muscle layer that generates weaker contractions [54]. This could have broad implications for research using gnotobiotic animals, which typically have larger ceca than SPF mice, potentially leading to inhomogeneous distributions of bacterial composition and functions. Future research in the field should therefore consider the heterogeneity in microbial and nutrient composition within the cecum when collecting and analyzing samples, and precise sampling locations should be reported.

Spatial heterogeneity in the gut on a microscopic scale due to biogeographical features, such as proximity to the epithelium or residence in crypts, has been described before [8–10,25], but we found that clustering of bacteria of the same species occurs in the lumen and does not require proximity to the epithelium. We demonstrated that this can occur independently of antibody-mediated cross-linking and active swimming of cells toward each other and hypothesize that these clusters likely originate from microbial growth in conjunction with the rheological properties of cecum content: without active mixing, cecum content behaves as a thick gel, holding bacterial offspring in place after cell division. Qualitatively similar microcolony formation has been observed for bacteria such *Staphylococcus aureus* [55] or *Vibrio cholerae* [28] entrapped in agarose gels. While hydrogels and gut content likely differ substantially in their underlying physicochemical

properties, both systems can impose physical constraints that limit dispersal, allowing microbial growth to result in cluster formation. Research in zebrafish has also described clustering of gut bacteria and suggested that gut contractions might play a role in limiting cluster size [25]. We have now demonstrated this in mice and found evidence that human gut content has rheological properties that could allow the same phenomenon in the human colon. Formation of bacterial clusters in the intestinal tracts of animals that is driven by growth and opposed by gut contractions might therefore be a general phenomenon.

The statistical method we use to investigate bacterial distribution in gut content, Ripley's H function, is a good tool to qualitatively understand whether bacteria are clustered, randomly distributed, or regularly spaced (Fig 2A). While the absolute value of H(r) is a function of the number of bacterial cells found in a given radius around a focal bacterium, due to the normalization by the radius and by the overall density of bacteria on the image (see Materials and methods), a quantitative interpretation of this value is complex [56]. It is, however, possible to extract meaningful quantitative information on cluster size from the H(r) plots, and it has been shown that the radius where H(r) reaches its maximum value approximates the real radius of a cluster within a factor of 2 [37]. In our analysis, these radii are generally <10 μm, indicating aggregates consisting of only few cells. This small cluster size fits the proposed growth-based mechanism for cluster formation, even though little data is available on bacterial growth in the gut lumen and the exact timing of gut contractions under physiological conditions.

When testing ricionleic acid and loperamide as targeted interventions to modify spatial structure in the microbiota, we obtained surprising results. As expected, ricinoleic acid, a compound that actuates smooth muscle contractions, led to less clustering of bacteria in the cecum of 3MM mice. However, loperamide, a drug that inhibits contractile activity, had no effect 1.5 h after treatment, and diminished clustering 6 h after initial treatment. Given our proposed growth-based mechanism for clustering, it is possible to explain the lack of a short-term effect of loperamide treatment; the time after treatment might simply not be long enough for bacterial growth to form clusters. The effect of less microbial clustering despite less contractile activity is harder to rationalize. This effect is weaker in the cecum tip and stronger in the base, close to inflow from the ileum and outflow to the colon, which could indicate that flow dynamics might play a role. For example, obstructed outflow could lead to more local content flow in the cecum. Loperamide treatment has also been associated with less food intake [49], which could lead to less nutrient availability for the microbes, diminishing growth. While these results highlight the intricate physiological interplay governing the host-microbiota system, they also show that pharmacological interventions can serve as a lever to change spatial structure in the microbiota, even though further work is needed to mechanistically understand how they work.

Bacterial clustering at the micro-scale can have functional consequences. Metabolic interactions between bacteria happen at very short ranges [12,57], and spatial arrangement thus plays a crucial role for allowing or blocking ecological crosstalk between bacterial cells. In addition, the duration of spatial assortment can affect whether interactions between cells can play a role in evolutionary processes. Gut contractions which disperse clonal microcolonies can therefore play an important role in maintaining or disrupting interspecies interactions, giving the host a direct way to shape ecological and evolutionary processes in the gut microbiota. Micrometer-scale clustering, similar to what we observe in the mouse cecum, has been shown to lead to changes in antibiotic susceptibility in nutrient replete conditions in vitro [55], and gel-entrapped microbes generally are known do exhibit similarly reduced susceptibility to treatment with antimicrobials as bacterial cells growing in biofilms [58–61].

Our study highlights the importance of quantitatively understanding the spatiotemporal dynamics of bacterial activity in the gut microbiota. Using image analysis in combination with statistics on spatial point patterns, we found that bacteria form microcolonies in the gut. By investigating how microbial behavior is affected by the physical properties of its environment, we were able to determine the relative importance of various factors that might contribute to this phenomenon. We conclude that bacterial microcolony formation is largely influenced by two competing properties, bacterial growth and gut contractions, which likely play an important role in gut microbial ecology.

## Lead contact

Further information and requests for resources and reagents should be directed to and will be fulfilled by the lead contacts, Markus Arnoldini (markus.arnoldini@hest.ethz.ch) or Giorgia Greter (giorgia.greter@hest.ethz.ch).

## Materials and methods

### Ethics statement

Experiments using mice were approved by the Swiss Kantonal authorities (Animal experimentation licenses ZH120/19 and ZH016/21). Human samples were collected during the NBMISI study (ClinicalTrials.gov ID: NCT04978077), which operates with the approval of the responsible local ethics committee (KEK Bern, project ID 2021-01108, approved on October 20th, 2021). Only patients who provided written informed consent were included in the NBMISI study, and samples were analyzed anonymously.

### Experimental design

The research objective of this study was to determine the causes of local spatial structure in the murine cecum and estimate the extent of microbial spatial structure in the human large intestine. The overall design used controlled laboratory experiments and samples of human ileal effluent estimate parameters of physical properties of the human intestine.

Wild type C57BL/6 mice were maintained on a standard chow diet at the ETH Phenomics center. Mice were re-derived Germ-free or bred with the 3MM or OligoMM[12] microbiotas and kept in isolators. The 3MM microbiota was derived using strains *Bacteroides thetaiotaomicron Δtdk* [62] (a strain that behaves like wild type, but allows counterselection for genetic engineering, derived from ATCC29148), *Eubacterium rectale* ATCC33656, and *Escherichia coli* HS [63]. SPF mice were bred and housed in individually ventilated cages. All experiments began at 8–14 weeks of age. Mice were fed *ad libitum* for the duration of all experiments and were kept with a dark and light period of 12 h each.

For experiments in which mice were colonized with bacteria, an endpoint was selected of 2 days post colonization, allowing bacteria to reach a steady state colonization levels in the cecum.

### DNA isolation and 16S sequencing

Cecum content was collected from 3MM, OligoMM[12], and SPF mice at the tip and the base. Bacterial DNA was extracted using the DNeasy PowerSoil Pro Kit 420 (Qiagen, Germany). 16S DNA sequencing was performed with Biomarkers Technologies (BMKGENE). Illumina sequencing was performed using an Illumina NovaSeq 6000, with paired end reads and read length of 100,000 (Illumina).

The raw sequencing data were analyzed using DADA2 v1.14 [64]. Specifically, primer sequences were removed with cutadapt v2.8 (515F = GTGCCAGCMGCCGCGGTAA, 806R = GGACTACHVHHHTWTCTAAT) and only inserts that contained both primers and were at least 75 bases long were kept for downstream analysis [65]. Next, reads were quality filtered using the filterAndTrim function of the dada2 package (maxEE = 2, truncQ = 3, trimRight = [40]). The learnErrors and dada functions were used to calculate sample inference using pool = pseudo as parameter. Reads were merged using the mergePairs function and bimeras were removed with the removeBimeraDenovo (method = pooled). Remaining amplicon sequence variants were then taxonomically annotated using the IDTAXA classifier in combination with the Silva v138 database [66,67].

### Calculating Jensen-Shannon divergence between microbiomes

For each sample, the relative species abundance based on 16S sequencing data can be interpreted as a probability distribution on the set of bacteria. The Jensen-Shannon divergence is a way to compare two probability distributions, with low/

high values indicating similarity/dissimilarity. If $B$ is the set of bacterial strains, and $P$ and $Q$ are two distributions on $B$, the Jensen-Shannon divergence between $P$ and $Q$ is defined as

$$JSD(P, Q) = \frac{1}{2}D_{KL}\left(P \parallel M\right) + \frac{1}{2}D_{KL}\left(Q \parallel M\right),$$

where $M = \frac{1}{2}(P + Q)$ is a mixture distribution of $P$ and $Q$, and $D_{KL}$ is the Kullback–Leibler divergence defined as

$$D_{KL}\left(P \parallel M\right) = \sum_{b \in B} P(b) log\left(\frac{P(b)}{M(b)}\right)$$

While the Kullback–Leibler divergence is asymmetric, the Jensen-Shannon divergence is symmetric. We computed the Jensen-Shannon divergence for each sample pair, with results shown in S2C and S2D Fig. Samples were categorized into base samples and tip samples. We then aggregated the Jensen-Shannon divergence values from each base sample to all other base samples (forming the base/base group) and from each tip sample to all other tip samples (forming the tip/tip group). Additionally, we calculated the divergence from base samples to tip samples, creating the base/tip group. A one-sided Mann–Whitney $U$ test was conducted to assess differences in distances between these groups, as illustrated in S2E and S2F Fig. For instance, in SPF mice, base samples exhibited greater similarity to one another compared to the tip samples, a trend not observed in OligoMM[12] mice.

## Fluorescent in-situ hybridization

Following euthanasia, the entire mouse cecum was harvested for microscopy. Freshly harvested samples were submerged in 4% paraformaldehyde and incubated overnight at 4 °C. Samples were then transferred to a 20% sucrose solution and incubated for 4 h at room temperature. After fixation, the samples were cut into tip and base sections and embedded in O.C.T (Tissue-Tek, O.C.T Compound), snap frozen in liquid nitrogen and stored at −80 °C. Cecum tip and base were cut into 5 µM thick sections using a Microm HM525 cryotome (ThermoFischer Scientific). The sections were mounted on a Superfrost++ glass slides and stored at −20 °C for further use.

FISH was used to stain each 3MM member. Dried tissue sections were contoured with a hydrophobic pen (ImmEdgeTM, Vector laboratories). Subsequently, samples were washed with PBS, dehydrated with increasing concentrations of ethanol (50%, 80%, 100%, vol/vol) and air dried. Five probes (one for *E. rectale,* one for *E. coli,* and three for *B. theta*) were used to detect the 3 bacterial species and were diluted in the hybridization buffer (for 100 ml: 10 ml 5M NaCl, 2 ml 1M Tris-HCl, 0.1 ml 10% SDS, 10 ml formamide, 69.9 ml $H_2O$) to a final concentration of 1% (vol/vol) each. The probe mix was added to the sections and incubated for 4 h in a dark moist chamber at 50 °C. Washing buffer (for 100 ml: 10 mL 5M NaCl, 2 ml 1M Tris-HCl, 0.1 ml 10% SDS, 79.9 ml $H_2O$) was added to the sections and incubated for 20 min at 50 °C. Three additional washing steps of 10 min were performed with PBS. After washing, the slides were rinsed with ice cold ddH2O, air dried and mounted with Vectashield H-1400TM Hardset Antifade mounting medium (Vector Laboratories). Slides were stored at −20 °C for further use.

## Niche size experiment

To evaluate the effect of niche size on bacterial clustering, an mCherry fluorescent *B. theta* strain was used [68] to colonize OligoMM[12] mice. Mice were colonized for 3 days and subsequently euthanized. Entire cecum was collected, fixed, cut into sections, and mounted as described above.

## Motility experiment

To evaluate the effect of bacterial motility on bacterial clustering, *E. coli* Z1331 wild type and mutant strains were used [69,70]. Gene deletions (*cheY*::*aphT* or *flhD*::*aphT*) in *E. coli* were obtained via PCR-based inactivation of chromosomal

genes, as previously described [71]. Ampicillin resistance and *mCherry* fluorescence genes were inserted using plasmids pSIM5 and pFPV25.5 mCherry [51]. Germ-free mice were colonized with *E. coli, E. coli cheY::aphT,* or *E. coli flhD::aphT*. Mice were colonized for 3 days and subsequently euthanized. Entire cecum was collected, fixed, cut into sections, and mounted as described.

## Particle mixing experiment

To evaluate whether small particles are mixed in the mouse cecum, 6, 3MM mice were treated by oral gavage with 100 µL of 10⁹ Fluoresbrite Yellow Green 1.0 µM Microspheres (Polysciences). Mice were euthanized 4.5 h after treatment and entire cecum was collected, fixed, cut into sections, and mounted as described.

To determine the flow of small particles in the mouse cecum, SPF mice were treated by oral gavage three times, 40 min apart with 50 µL of 10⁹ Microspheres. The first treatment was with Fluoresbrite Polychromatic red 1.0 µM Microspheres (Polysciences), followed by Fluoresbrite Polyfluor 407 1.0 µM Microspheres (Polysciences), and finally with Fluoresbrite Yellow Green 1.0 µM Microspheres (Polysciences). Mice were euthanized 5 h after the first treatment with polychromatic red microspheres and entire cecum was collected, fixed, cut into sections, and mounted as described.

## Manipulating gut contractions

To evaluate the effect of gut contractions on bacterial clustering, 3MM mice were treated with ricinoleic acid or PBS to increase gut contractions or as a negative control, respectively. Mice were injected intraperitoneally with 100 µL of PBS, 100 µl PBS containing 8 mg of ricinoleic acid (previously filtered through a 0.22 µm filter, based on [47]), or 0.4 mg loperamide in 100 µl PBS, and euthanized after 90 min. For the long-term effect of loperamide, mice were injected intraperitoneally with 0.035 mg loperamide dissolved in 0.9% NaCl at 0 h, and again with the same concentration of loperamide at 2 h, and subsequently sacrificed at 6 h. The cecum was fixed, sectioned, stained using FISH, and mounted as described.

## Image acquisition and processing

Acquisition of bacterial images in the cecum was performed using a Leica TCS SP8 STED confocal microscope under a HC PL APO CS2 63x/1.4 oil immersion objective. The *E. coli* probe, tagged with an Atto 425 fluorophore, was excited with a 458 nm argon laser and emission signals were detected between 471/539 nm. A DPSS 640 nm laser was used to excite the *B. theta* probes and a DPSS 471 nm laser to excite the *E. rectale* probe. The emission was detected in a 650/712 nm and 576/618 nm window, respectively. Sections of cecum containing *mCherry* fluorescent bacteria were excited with a 458 nm argon laser and emission signals were detected between 471/539 nm.

For fluorescent beads in particle mixing experiments, Fluoresbrite Polyfluor 407 1.0 µM Microspheres were excited with a 405 nm argon laser and emission signals were detected between 415/440 nm. Fluoresbrite Yellow Green 1.0 µM Microspheres were excited with a 458 nm argon laser and emission signals were detected between 468/500 nm. Fluoresbrite Polychromatic red 1.0 µM Microspheres were excited with a 514 nm argon laser and emission signals were detected between 524/600 nm.

Images were acquired with a resolution of 2048 × 2048 pixels. Ten Images were acquired as Z-stacks across the 5 µM thick section. After acquisition, images were deconvoluted using the Huygens Remote Manager. A signal to noise ratio of 9 was used for the deconvolution of *E. coli*, and signal to noise ratio of 17 was used for the deconvolution *E. rectale* and *B. theta*. The deconvolution algorithm "cmle" was used in 30 iterations. The resulting deconvoluted stacks were compressed into one layer using a maximum intensity projection in ImageJ (v 1.51n) and each channel image was stored in a separate folder.

Four images of each mouse cecum section were taken, two at the epithelial border and two in the center of the cecum. The images covered a total area of 184.5 µm². For the experiment evaluating fluorescent bead mixing in 3MM mice, the resolution was decreased to 1024 × 1024 pixels to increase image acquisition speed.

For the experiments evaluating *E. coli* mixing through IgA-mediated clustering in 4-week colonized mice, as well as for evaluating small particle flow direction in the cecum in SPF mice, two images of each mouse cecum section were taken: one at the epithelial border and one in the center of the cecum. Epithelial images were composed of 6 square images stitched into a line and center images were acquired as 9 single images stitched into a larger square. Epithelium images covered a total area of ~85 × 510 μm and center images an area of ~255 μm².

## Image analysis

An image analysis pipeline was developed in Matlab (R2022a) to identify the coordinates of each bacterium in microscopy images. Briefly, image preprocessing was done using top hat filtering and local maxima adjustments. A binarization threshold was set based on the average pixel intensity in each image and single cells were identified using size, eccentricity, and solidity thresholds. Food particles were removed from the images based on size and length/width ratios. Cells in clusters were identified two times by edge detection and subsequent dilation/erosion. The total cell count per image, coordinates, and area of each bacterium in the image was stored for further analysis.

## H function

Ripley's H function is a transformation of Ripley's K-function, in which the number of neighbors of bacteria are calculated at different radii (r) from all cells and compared to simulations of complete spatial randomness. Since the images show considerable local heterogeneity in the density and distribution of cells, possibly due to food particles, we used the inhomogeneous H function, rather than its homogeneous version, because it accounts for local density variation: the inhomogeneity of bacterial density is accounted for by weighing each point to its local density ($K_{inhom}$). The formula used in Ripley's inhomogenous K-function is as follows:

$$K_{inhom}(r) = \mathbb{E}[\sum_{x_j \epsilon \boldsymbol{X}} \frac{1}{\lambda(x_j)} \mathbf{1} \{0 < ||u - x_j|| \leq r\} |u \epsilon \boldsymbol{X}],$$

defined as the expected number of neighboring points $x_j$ within a radius $r$ of a focal point $u$ of the point process $\boldsymbol{X}$, weighted by the local density at that point $\lambda(x_j)$.

The L-function transforms the K-function such that images with point patterns of complete spatial randomness will have $L_{inhom}(r) = r$ for all distance bands, while images with clustered bacteria will have $L_{inhom}(r) > r$, and images with regularly spaced bacteria will have $L_{inhom}(r) < r$. The L-function relates to the K-function as follows:

$$L_{inhom}(r) = \sqrt{\frac{K_{inhom}(r)}{\pi}}$$

The H function is a normalization of $L_{inhom}(r)$, such that all values of complete spatial randomness would be $H_{inhom}(r) = L_{inhom}(r) - r = 0$. All images with clustered bacteria would be $H_{inhom}(r) > 0$, and all images with regularly distributed bacteria would be $H_{inhom}(r) < 0$. Using this transformation, we can visualize at which radius clustering is occurring based on the curve's deviations from 0.

To test whether the $H_{inhom}(r)$ curves were significantly different from each other, a Studentized permutation test, specifically developed for this application [41,72] and implemented in the R-package "spatstat", was carried out with 1,000 permutations over an analysis radius of 0–20 μm.

Border effects on the images were corrected for using the built-in border-correction function in the spatstat package. We consistently used the method 'border', which excludes points which are closer to the border than the radius r. However, the various border-correction methods implemented in spatstat produced qualitatively similar output.

For calculating H(r) functions, plotting, and associated statistical analysis, the R-package "spatstat" V4.3.2 [73] was used. Specifically, we employed the *Linhom* function with the parameters *correction = c("border"), renormalize = FALSE,* and *ratio = TRUE*, while maintaining the standard settings for all other options.

## IgA measurements

Wild type GF mice were colonized with the 3MM microbiota for 4 weeks. Intestinal lavage was collected in 2 mL of PBS from ex-GF mice as well as from WT mice bred with the 3MM microbiota and stored at −80 °C for further analysis. A flow cytometry technique was applied to estimate 3MM-specific IgA responses [74]. Intestinal lavages were diluted 5 times in 3 folds across a 96-well plate, starting from undiluted samples. The 3MM bacteria were grown overnight in filtered BHIS, washed, and quantified by flow cytometry. Subsequently they were diluted in PBS containing 2% BSA and 0.02% $NaN_3$ to a concentration of roughly $10^5$ bacteria and added to each intestinal lavage dilution. BrilliantViolet 421 anti-rat IgA antibody (BD biosciences) was used to detect IgA. Plates were read by a CytoFlex Flow Cytometer (Beckman Coulter). Bacteria were identified by light scattering methods and IgA bound to bacteria by median fluorescent intensity (MFI) using FlowJo V10.8 Software (BD Life Sciences).

## Rheology of human samples

Stoma effluent samples were collected from 3 ileostomy patients of the NBMISI (Networks of bacterium-metabolite interactions in the small intestine) study. Samples were taken by the study physician with a sterile syringe directly from the stoma bag. Samples at time point 0 h are collected after an overnight fasting period. Subsequent timepoints indicate collection times after a defined nutritional intervention (Nutricia Calogen or Nutricia preOp, respectively after an overnight fasting period); e.g., time point 3 h indicates sampling at 3 h after ingestion of a nutritional intervention. No further food was ingested after the nutritional intervention. Water intake was assessed but not restricted after the nutritional interventions.

## Rheology

To measure the rheological properties of mouse cecum content, the entire cecum content was harvested from 3MM mice and stored at −80 °C for further analysis unless otherwise noted. Human ileostomy samples were also stored at −80 °C prior to rheological measurements were performed.

All rheological measurements were conducted using a stress-controlled rheometer (Anton Paar MCR302) equipped with a parallel-plate geometry (diameter d = 25 mm, constant gap h = 0.8 mm). To minimize wall slip, both plates were covered with grit sandpaper (P120) using double sided tape. Temperature for all measurements was set to 37 °C using a Peltier temperature control system and the system was covered with a protective hood.

For oscillatory measurements, sample drying was prevented by adding a wet sponge to the inside of the rheometer chamber. Strain amplitude sweeps were performed at a fixed angular frequency of $\omega = 1 \frac{rad}{s}$, with the strain aplictude ramped from $\gamma = 0.1$ *to* 1000%. Frequency sweeps were performed at a fixed strain of $\gamma = 0.1$% and over an angular frequency range of $\omega = 0.1$ *to* $100 \frac{rad}{s}$.

For creep measurements, which require longer measurement times, sample drying was prevented by immersing the sample edge in a bath of light mineral oil (viscosity η = 10 mPa s at 40 °C). After loading, samples were first conditioned by a strain amplitude sweep, as described above, followed by a 5 min time sweep, during which the temporal evolution of the viscoelastic properties was monitored at a constant strain of $\gamma = 0.1$% and frequency $\omega = 1 \frac{rad}{s}$. Creep tests were then performed by applying constant shear stresses ranging from 10 to 50 Pa in increasing order, each applied for 300 s

and separated by a recovery period of 100 s. The yield stress was defined as the applied stress at which the shear rate increased continuously with time, indicating material fluidization.

## Contraction depth modeling

We modeled the gut wall contractions and the corresponding mixing process as the 'Stokes second problem', a classic fluid dynamics problem, whereby the wall contractions are represented as an oscillating plane (in-plane oscillations) that mixes the fluid (digesta) above it. We approximated the digesta as a Newtonian fluid of density $\rho$ and dynamic viscosity $\mu$. Digesta density $\rho$ was determined by measuring weight and volume changes simultaneously when digesta was added to water (results: 1.16 g/ml, 1.29 g/ml, 1.31 g/ml, for three independent replicates). The rigid plane is located at $y = 0$ and oscillates with velocity $A\omega \cos(\omega t)$ along the $x$-direction, where $A$ and $\omega$ are the amplitude and frequency of the oscillations. The digesta velocity profile then reads

$$u(y, t) = A\omega \exp\left(-\sqrt{\frac{\omega}{2\nu}}y\right) \cos(\omega t - \sqrt{\frac{\omega}{2\nu}}y),$$

where $\nu = \mu/\rho$ is the kinematic viscosity of the digesta. The shear stress is

$$\tau(y, t) = \mu\frac{\partial u}{\partial y} = A\omega\sqrt{\rho\omega\mu} \exp\left(-\sqrt{\frac{\omega}{2\nu}}y\right) \sin\left(\omega t - \sqrt{\frac{\omega}{2\nu}}y - \pi/4\right).$$

We first compute the shear stress at $y = 0$, which represents the force per unit area the gut wall exerts on the digesta

$$\tau(y = 0, t) = A\omega\sqrt{\rho\omega\mu} \sin(\omega t - \pi/4).$$

The amplitude of the force is thus $A\omega\sqrt{\rho\omega\mu}$, which is plotted in S7 Fig as a function of the amplitude $A$ for various oscillation frequencies $\omega$.

To estimate the mixing depth $D$, we match the amplitude of the viscous shear stress at $y = D$, with the yield stress $\tau_{yield}$

$$A\omega\sqrt{\rho\omega\mu} \exp\left(-\sqrt{\frac{\omega}{2\nu}}D\right) = \tau_{yield}.$$

Solving for $D$, gives

$$D = -\sqrt{\frac{2\nu}{\omega}}\log\left[\tau_{yield}/(A\omega\sqrt{\rho\omega\mu})\right].$$

For producing the plots in Fig 4B, we estimated the following parameters to solve for D (Table 1):

**Table 1. Parameters used to model the effect of contractions on gut content.**

| Parameter | Value | Source |
|---|---|---|
| Digesta yield stress $\tau_{yield}$ | 45 Pa | rheology, this study |
| Digesta dynamic viscosity $\mu$ | 2,237 Pa s | rheology, this study |
| Digesta density $\rho$ | 1,250 kg/m³ | density measurements, this study |

## Supporting information

**S1 Fig. Example images and image analysis/cell detection using the image analysis pipeline developed for this study.** (**A**) Representative image of the cecum tip, center, of a 3MM mouse. (**B**) Left column shows confocal microscopy images for the three bacterial species in the 3MM microbiota, detected by FISH probes in fixed cecum content. Right column shows the binarized images resulting from image analysis. The data underlying this Figure can be found in the data repository accompanying this paper under the DOI: https://doi.org/10.5281/zenodo.19422141.
(TIF)

**S2 Fig. The microbiota is not well mixed in the cecum of SPF and OligoMM¹² mice.** (**A**) Results of 16S sequencing of cecum content of OligoMM¹² ($n = 3$) and (**B**) SPF mice ($n = 5$) at the family-level. Tip and base samples of the same cecum were sequenced separately. (**C**) Jensen-Shannon distances between each sequenced tip and base for OligoMM¹² and (**D**) SPF mice at the family-level. (**E**) Plots of base/base, base/tip, and tip/tip Jensen-Shannon distances across all mice in the OligoMM¹² group and (**F**) the SPF group. Significance was tested with a one-sided Mann–Whitney $U$ test. The data underlying this Figure can be found in the data repository accompanying this paper under the DOI: https://doi.org/10.5281/zenodo.19422141.
(TIF)

**S3 Fig. Alternative representation of the data shown in in** Fig 1E. (**A**) Bead count in cecum tip (blue) an cecum base (orange) for beads administered 220 min (green beads), 260 min (blue beads), and 300 min (red beads) before sacrifice. Insert shows a schematic representation of the cecum, highlighting location of tip and base. (**B**) Bead count in the cecum tip, in images taken close to the epithelium (purple) or in the center (blue) of the cecum, depending on the time after bead administration. (**C**) Bead count in the cecum base, in images taken close to the epithelium (purple) or in the center (blue) of the cecum, depending on the time after bead administration. The insert in (**A**) was created using BioRender (https://biorender.com/58n1m7d). The data underlying this Figure can be found in the data repository accompanying this paper under the DOI: https://doi.org/10.5281/zenodo.19422141.
(TIF)

**S4 Fig. 3MM microbiota members, but not beads, cluster in the cecum lumen.** Results for all three 3MM species when cell distribution was analyzed using the inhomogeneous H function. Shaded regions indicate 95% CI ($n = 6$). Red dashed lines indicate the theoretical value for complete spatial randomness. Locations are (**A**) base, epithelium, (**B**) base, center, (**C**) tip, epithelium, and (**D**) tip, center. H(r) functions for all bacterial species are significantly different from complete spatial randomness ($p < 0.01$). With the exception of *E. rectale* at the epithelium of the cecum base, H(r) functions for all bacteria data were significantly different from H(r) function for beads ($p < 0.01$). H(r) functions for beads are significantly different from complete spatial randomness at the base epithelium ($p = 0.04$), but not at the base center. No beads were found at the cecum tip. Bonferroni-corrected Studentized permutation tests were used for statistical comparisons between H(r) functions. The data underlying this Figure can be found in the data repository accompanying this paper under the DOI: https://doi.org/10.5281/zenodo.19422141.
(TIF)

**S5 Fig. Effects of sIgA on bacterial colonization and visual cluster formation.** (**A**) Cell counts, based on image analysis, for *E. coli* in mice that were colonized from birth with the 3MM microbiota (blue), and mice that were colonized for 4 weeks (red). Only mice that were colonized for 4 weeks have sIgA antibodies against *E. coli*. (**B**) Results for spatial distribution of wt (blue), Δ*cheY* mutant (yellow), and Δ*flhD* mutant *E. coli* cells in the cecum ex-germ-free mice, using the inhomogeneous H function ($n = 12$, 4 in each group, cecum base epithelium data shown). All groups show significant clustering

($p < 0.01$), and there is no significant difference in clustering between wild type and $\Delta cheY$ ($p > 0.05$), but between wild type and $\Delta flhD$ ($p = 0.12$). The form of the H(r) function suggests that $\Delta flhD$ clusters more than wild type, indicating that flagella might be important for cluster dispersal rather than cluster formation. Bonferroni-corrected Studentized permutation tests were used for statistical comparison between H(r) functions. The data underlying this Figure can be found in the data repository accompanying this paper under the DOI: https://doi.org/10.5281/zenodo.19422141.
(TIF)

**S6 Fig. Rheology measurements to determine the yield stress of gut content in mice and humans.** (**A**) Shear stress as a function of shear strain in five samples of 3MM mouse cecum content in oscillatory measurements (same dataset as shown in Fig 4A). The lines are power-law fits of the behavior well above (full lines) and below (dashed lines) the yield point, and the Y coordinate of their intersection points corresponds to the yield limit (yield stress values: 47.79 Pa, 50.13 Pa, 39.66 Pa, 36.25 Pa, 58.21 Pa, for samples 1–5, respectively). (**B**) Same analysis as (**A**), but for four samples of human ileal effluent (same dataset as shown in Fig 4E). Yield stress values are 0.03 Pa, 0.44 Pa, 0.50 Pa, 0.03 Pa, for samples 1–4, respectively. (**C**) Creep compliance as a function of time for three independent samples of 3MM mouse cecum content. The arrows indicate the shear stress where creep compliance increases linearly with time (indicating fluidization of the material), the yield limit. The data underlying this Figure can be found in the data repository accompanying this paper under the DOI: https://doi.org/10.5281/zenodo.19422141.
(TIF)

**S7 Fig. Dependence of shear stress experienced by gut content on the amplitude and frequency of gut contractions.** The shown results are based on Stokes' second problem and take measured values for viscosity and density of gut content into account. Dashed line indicates measured yield stress of gut content (45 Pa). The data underlying this Figure can be found in the data repository accompanying this paper under the DOI: https://doi.org/10.5281/zenodo.19422141.
(TIF)

**S8 Fig. Analysis of population sizes and clustering under pharmacologically modified gut contractions.** (**A**) Cell counts on microscopy images in ricinoleic acid and PBS treated groups. In most cases (all except *E. rectale* base center) where we have data for all 4 treatment groups, significant differences exist in cell counts between groups (ANOVA, **, ***, **** correspond to $p < 0.01$, 0.001, and 0.0001, respectively) (**B**) Clustering analysis using the inhomogeneous H function for all three 3MM strains after IP injection of PBS (red), ricinoleic acid (cyan), and loperamide (purple) 1.5 h after treatment, and for loperamide 6 h after treatment (green), for different locations in the cecum. Studentized permutation tests were used to assess whether curves were significantly different, and $p$ values were Bonferroni-corrected for the number of comparisons (3 in each case: control versus ricinoleic acid, control versus loperamide 1.5 h, control versus loperamide 6 h). For *E. coli*, the following curves were significantly different: control versus loperamide 6 h (base center; $p = 0.003$), control versus loperamide 6 h (base epithelium; $p = 0.006$), control versus loperamide 6 h (tip center; $p = 0.006$). For *B. theta*, we found significant differences between the following curves: control versus loperamide 6 h (base center; $p = 0.003$), control versus loperamide 6 h (base epithelium; $p = 0.003$), control versus loperamide 6 h (tip center, $p = 0.003$), control versus loperamide 6 h (tip epithelium, $p = 0.024$). For *E. rectale*, the following comparisons differed significantly: control versus loperamide 6 h (base center; $p = 0.003$), control versus loperamide 6 h (base epithelium; $p = 0.003$), control versus loperamide 6 h (tip epithelium, $p = 0.039$). Of note, the comparisons between control and ricinoleic acid that were significantly different with $p < 0.05$ without Bonferroni correction (*E. coli*: tip epithelium, $p = 0.02$; *B. theta*: tip epithelium, $p = 0.023$) lost that significance level after multiple testing correction for three tests. The data underlying this Figure can be found in the data repository accompanying this paper under the DOI: https://doi.org/10.5281/zenodo.19422141.
(TIF)

## Acknowledgments

We would like to acknowledge Jan Vermant and Jonas Cremer for critical reading of the manuscript, and all members of the laboratory for Mucosal Immunology at ETH Zürich for their input. We thank Sven Nowok and Dominik Bacovcin, as well as all the animal caretakers at the EPIC facility at ETH Zürich for maintaining the mouse lines and for their experimental support.

## Author contributions

**Conceptualization:** Giorgia Greter, Suwannee Ganguillet, Eleonora Secchi, Emma Slack, Markus Arnoldini.

**Data curation:** Giorgia Greter, Sebastian Hummel.

**Formal analysis:** Giorgia Greter, Sebastian Hummel, Daria Künzli, Suwannee Ganguillet, Milad Radiom, Claudia Moresi, Steffen Geisel, Julien Bauland, Jonasz Slomka, Eleonora Secchi, Markus Arnoldini.

**Funding acquisition:** Markus Arnoldini.

**Investigation:** Giorgia Greter, Daria Künzli, Naomi Dünki, Niina Ruoho, Patricia Burkhardt, Suwannee Ganguillet, Claudia Moresi, Leanid Laganenka, Steffen Geisel, Julien Bauland, Sebastian Jordi, Benjamin Misselwitz, Bahtiyar Yilmaz, Jonasz Slomka.

**Methodology:** Giorgia Greter, Sebastian Hummel, Naomi Dünki, Suwannee Ganguillet, Claudia Moresi, Steffen Geisel, Julien Bauland, Sebastian Jordi, Benjamin Misselwitz, Bahtiyar Yilmaz, Jonasz Slomka, Eleonora Secchi, Emma Slack, Markus Arnoldini.

**Project administration:** Emma Slack, Markus Arnoldini.

**Resources:** Suwannee Ganguillet, Leanid Laganenka, Wolf-Dietrich Hardt, Sebastian Jordi, Benjamin Misselwitz, Bahtiyar Yilmaz, Roman Stocker, Emma Slack.

**Software:** Giorgia Greter, Sebastian Hummel, Naomi Dünki.

**Supervision:** Emma Slack, Markus Arnoldini.

**Validation:** Giorgia Greter.

**Visualization:** Giorgia Greter, Sebastian Hummel.

**Writing – original draft:** Giorgia Greter, Markus Arnoldini.

**Writing – review & editing:** Giorgia Greter, Sebastian Hummel, Milad Radiom, Wolf-Dietrich Hardt, Jonasz Slomka, Eleonora Secchi, Roman Stocker, Emma Slack, Markus Arnoldini.

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
