## [Editor Report · Decision Letter 0]

28 May 2025

Dear Dr Arnoldini,

Thank you for submitting your manuscript entitled "Emergent spatial structure in the gut microbiota is driven by bacterial growth and gut contractions" for consideration as a Research Article by PLOS Biology.

Your manuscript has now been evaluated by the PLOS Biology editorial staff, as well as by an academic editor with relevant expertise, and I am writing to let you know that we would like to send your submission out for external peer review.

Once your full submission is complete, your paper will undergo a series of checks in preparation for peer review. After your manuscript has passed the checks it will be sent out for review. To provide the metadata for your submission, please Login to Editorial Manager (https://www.editorialmanager.com/pbiology) within two working days, i.e. by May 30 2025 11:59PM.

Kind regards,

Melissa

Melissa Vazquez Hernandez, Ph.D.

Associate Editor

PLOS Biology

---

## [Decision Letter · Decision Letter 1]

15 Jul 2025

Dear Dr Arnoldini,

Thank you for your patience while your manuscript "Emergent spatial structure in the gut microbiota is driven by bacterial growth and gut contractions" was peer-reviewed at PLOS Biology. It has now been evaluated by the PLOS Biology editors, an Academic Editor with relevant expertise, and by three independent reviewers. Please accept my apologies for the long period for peer-review.

In light of the reviews, which you will find at the end of this email, we would like to invite you to revise the work to thoroughly address the reviewers' reports. While two reviewers are mostly positive, there are several important concerns raised. Reviewer 1 mentions that while the manuscript relies heavily on images, they provide too few of them. Reviewer 2 says that some of the conclusions are rather speculative, and requires further statistical analysis as well as further image-based analyses. As it is, the reviewer thinks that the role of digesta rheology still remains an open question and requires further experimental support. Reviewer 3 thinks the study may not reflect the biology of complex communities, and suggests additional experiments like testing the role of motility in clustering, further testing the model with other alterations like fiber content. We agree with all reviewer concerns and would require some additional experimental revisions to address them, as we consider that this would strengthen the work.

IMPORTANT: after discussion with the Academic Editor and the reviewers, we would strongly encourage the loperamide experiment suggested by Reviewer 3, which we think can also help in addressing some concerns of Reviewer 2.

Given the extent of revision needed, we cannot make a decision about publication until we have seen the revised manuscript and your response to the reviewers' comments. Your revised manuscript is likely to be sent for further evaluation by all or a subset of the reviewers.

**IMPORTANT - SUBMITTING YOUR REVISION**

*Re-submission Checklist*

*Published Peer Review*

*PLOS Data Policy*

*Blot and Gel Data Policy*

Sincerely,

Melissa

Melissa Vazquez Hernandez, Ph.D.

Associate Editor

PLOS Biology

REVIEWERS' COMMENTS

Reviewer #1:

In this lovely manuscript, Greter and colleagues leverage a diversity of experimental and computational techniques to explore multiple properties of the spatial organization of the murine gut microbiome and the molecular and mechanical forces that shape this organization. First, the authors use FISH to image the distribution of a model 3-member microbiota comprised of E. coli, B. thetaiotaomicron, and E. ractale. They use these images to precisely quantify the distribution of bacteria along the structure of the murine cecum, with a focus on regions near the center, near the epithelium, and near the cecal tip. They find a non-uniform cecal distribution which they then demonstrate is also present to some degree in both a specific pathogen free (SPF) microbiota as well as the OligoMM12 defined microbiota using sequencing of resected tissue. They then explore the role of peristaltic flow on the distributions by imaging the distribution of fluorescently labeled beads administered at 40 minute intervals within the mice and find that the beads support a order flow into the cecum. They then leverage geospatial statistical methods to characterize the clumpiness of bacteria versus beads and find that the bacteria have a greater degree of clumpiness—i.e. they are more likely to be found to bacteria of the same type than one would predict given the average density—and as an excellent control they show that this is not true for bead distributions. Next they explore a series of potential biological origins for this spatial clumping. They use a intriguing trick to deplete the mice of sIgA—specifically leveraging the fact that mice that were colonized with the three member community from birth do not generate bacterial reactive sIgA whereas germ-free mice that have this community administered do. Remarkably, they find a difference in the spatial distribution in two of the three members between these conditions. Thought they note that this is a relatively small difference, and they conclude that sIgA is not responsible for clumping. They also explore the role of motility, but find that in E. coli deficient for flagella production they still see similar distributions. To explore the role of growth, they modify the metabolic niche by exploring the distribution of B. theta in a more diverse microbiota—OligoMM12—with the idea being that with increased competition for nutrients, B. theta likely has a lower growth rate. They still see clustering, but the extent was more greatly reduced, suggesting that growth is indeed responsible for clustering. Finally, they investigate aspects of the cecal environment that could shape cluster size. They make microrheology measurements of cecal content and explore properties under perturbation rates that are likely similar to that seen during normal gut contraction. They then perturb contractions with a small molecule and show that this shifts the degree of clustering in E. coli.

Overall, this paper investigates a fascinating somewhat understudied topic, deploys a variety of methods, and thoroughly examines a series of plausible explanations for the patterns that are seen. I have no major concerns with this work and recommend publication.

The only minor concern that I would like to share with the authors is described below. I provide this only in the hope that it may help them strengthen an already strong submission.

My minor concern is that for a paper that relies so heavily on images of the spatial distribution of bacteria within the cecum, there are relatively few images provided. The Ripley's K based analysis (the H(r) function) is convincing, yet I think that many readers may share the same curiosity I had regarding the appearance of the actual spatial distributions. Outside of Figure 1B, 2C, S1, and S4B there are no images. Perhaps the authors would consider adding some additional supplemental figures with more example images of the distributions they see, including both replicates of the measurements provided as well as examples from the diversity of measurements analyzed but for which no images were shown.

Reviewer #2:

This paper investigates the mechanisms controlling the macro- and microscale organization of the microbiota in the murine caecum. The authors conclude that bacterial growth plays a major role in the formation of bacterial clusters (microcolonies) in the caecum, while gut contractions limit microcolony formation. The study is built upon a series of hypotheses introduced throughout the paper, which support this conclusion. However, the abstract does not clearly reflect the different hypotheses tested and places too much emphasis on the rheological properties of the digesta—an aspect that does not seem essential to explain the observed phenomena (see later). The experiments involve germ-free or gnotobiotic mice colonized with different microbiota. Bacterial localization is assessed on fixed caecal samples, and quantitative observations are performed using confocal microscopy.

Major comments

Some of the conclusions or interpretations appear somewhat speculative. I detail these points below.

In the Introduction and Discussion sections, the authors discuss how bacterial growth depends on physical heterogeneities in the gut. Given that this study focuses on the caecum, it would be relevant to describe the structural organization of the caecal mucosa (e.g., presence of crypts), as well as what is known about the mucus in this region of the gut (Ref. 47). Can we ignore the role of mucus in the caecum? Are there any oxygen gradients that could influence the microbiota?

A substantial part of the results relies on the interpretation of the Ripley's H function. While it is clear that the sign of H indicates whether bacteria are clustered, randomly distributed, or regularly spaced, can we assign a more quantitative meaning to the value of H? For example, if H at a given r is twice as high in one condition compared to another, does this carry any interpretable significance in terms of spatial organization? Consequently, is it meaningful to perform statistical comparisons on the H(r) profiles? How did the authors address boundary effects in their spatial analysis? Figure S1 appears to show considerable heterogeneity across the image (possibly due to food particles); could this influence the calculation of the H function?

Although this may not be straightforward, it would be highly informative to provide the probability distribution of aggregate sizes to support comparisons between experimental conditions. Such distributions could potentially be extracted via 2D FFT, image cross-correlation, or other image-based analyses.

A key result of the paper concerns the role of motility in the presence of aggregates. As currently presented, the authors suggest that the rheological properties of the digesta—specifically its yield stress / gel-like behavior—facilitate the breakdown of aggregates and their displacement from the epithelium to the center of the lumen. In the abstract, they state:

"In samples from mice and humans, we show that upper large-intestinal content behaves as a non-Newtonian fluid that changes its viscoelastic properties under the force of gut contractions. This phenomenon is sufficient to explain micro-scale bacterial clustering in the murine cecum, resulting from growth within the gel-like structure of cecum content, and periodic disruption due to peristalsis-driven shear-thinning and clearance."

This sentence implies that the physical properties of the digesta, in the absence of flow, are sufficient to promote bacterial growth in clusters—similar to what is observed in agarose gels (Ref. 50). However, I am doubtful about the validity of this comparison. Agarose is a crosslinked polysaccharide network that traps water, whereas digesta is a suspension of solid food particles in a liquid phase. It is therefore questionable whether digesta can be reasonably described as a "gel-like" matrix in the same sense as agarose. In digesta, bacteria may experience a more water-like environment and could potentially adhere to food particles or interfaces, which may play a more significant role in clustering. These considerations remain speculative and, in my opinion, should be clearly framed as such in the discussion. Furthermore, in classical Newtonian fluid flow driven by circular or longitudinal gut contractions, both velocity and shear stress are typically maximal near the wall. If the digesta behaves as a yield stress fluid, this effect may be attenuated, as the flow would occur primarily in regions where the stress exceeds the yield threshold. This rheological behavior might, in fact, protect bacteria located near the wall from being displaced. The authors' use of the Stokes' second problem to argue that oscillating contractions could overcome the yield stress and disrupt aggregates is potentially relevant, but remains a theoretical suggestion. In my view, Figure 4C convincingly supports the role of cecal motility in modulating bacterial clustering, but the role of digesta rheology remains an open question that would benefit from further experimental support.

Additionally, the authors state that:

"Shear-thinning can also explain the surprising observation that fed beads enter the tip of the mouse cecum by flow along the epithelial cell layer before being mixed into the cecum content."

This observation, however, is not unique to shear-thinning fluids. In flow within the gut, fluid motion is driven by boundary movement, leading to maximum shear and velocity near the wall, regardless of whether the fluid is Newtonian or non-Newtonian. This well-known feature of gut biomechanics has been previously described (e.g., see Figures 3, 8, and 9 in doi:10.1098/rsif.2013.0027).

Line 125:

Beads of different colors were administered at different time points. Figure 1D presents the relative abundance of beads after gavage, but the specific color coding is not described in the legend or figure. First, the histogram is somewhat difficult to interpret, as the evolution of each data ('tip center', etc.) is not clearly visible as in a standard plot. Second, additional details about bead colors and their corresponding time points would help clarify the experimental design. Have the authors attempted to plot the quantity of beads (by color/time) in each of the four anatomical segments? Such a representation could more directly support their conclusion that beads migrate along the epithelium before reaching the center of the lumen. Additionally, are these differences statistically significant?

Line 170:

How does the size of bacterial clusters compare to the size of a single bacterium? Is it possible to estimate the average number of bacteria per cluster? From the H(r) plots, could the authors extract or approximate a size distribution of aggregates? Given that many clusters appear to be composed of only 2-3 bacteria, this could inform whether the clusters arise from bacterial growth over time, and how this correlates with known bacterial replication rates. Such a quantitative link would strengthen the proposed growth-based mechanism of aggregation.

Line 278:

The conclusion that clusters represent clonal microcolonies is supported by the data. However, the reference to "transiently unmixed cecum content" does not appear to be directly supported by the results presented. Could the authors clarify whether there is experimental evidence supporting this statement?

Figure 4:

The description of Figure 4 in the main text does not correspondto what is actually shown in the sub-panels. The figure presents an amplitude sweep, in which both G′ and G′′ plateau at low strain amplitudes, supporting the gel-like behavior of the digesta. However, the reported yield stress value should be better substantiated. Ideally, it should be supported by creep tests (i.e., the time evolution of shear strain under a constant applied shear stress), which are more appropriate for identifying the onset of flow and determining the yield stress. I assume such tests may have been performed—if so, could the authors include them in the supplementary material to support the reported yield stress value?

Minor comments

* Lines 51-53: The sentence is unclear and should be rephrased for clarity.

* Lines 53-54: It would be helpful to indicate the typical values or characteristic ranges of short-range interactions between bacteria.

* Line 78: Seminal work by R.G. Lentle on digesta rheology should be cited alongside references 31 and 32. Suggested references include:

Lentle, R. G., & Janssen, P. W. M. (2008). Physical characteristics of digesta and their influence on flow and mixing in the mammalian intestine: a review. Journal of Comparative Physiology B, 178(6), 673-690.

* Line 264: The sentence is unclear and needs rephrasing for clarity.

* Line 395: The sentence appears contradictory: "In the cecum, dense mucus fills only the bottom of the intestinal crypts." Please clarify whether the mucus is localized or more diffusely distributed.

* Line 412: Could the authors clarify what is meant by "pre-existing structure"? Are they referring to structures in the physical environment (e.g., mucus, food particles) or to pre-formed bacterial colonies?

Reviewer #3:

In this manuscript, the authors examine the spatial structure of the mouse cecum using several complimentary approaches in 3 member, 12 member and SPF conventional communities as well as oral bead delivery. They find that cecal tip as compared to the base harbor different bacteria; these differences are more pronounced than the center to epithelial axis, which I found a bit surprising. The authors next observe that the bacteria are clustered and non-random in their distribution and go on to suggest that antibody binding and bacterial chemotaxis are not responsible for this. Finally, the authors demonstrate that smooth muscle contraction/motility can influence the distribution of clusters in a non-random fashion by augmenting function with ricinoleic acid to show decreased clustering. Overall, the manuscript contains high quality data and interesting findings. The manuscript would be appropriate for this journal with some minor revisions.

Strengths

-use of appropriate model systems and technology to clearly assess their questions

-A logical line of questioning with insights into physical properties influencing cecal microbiome spatial dynamics

-inclusion of human ileal effluent samples

Weaknesses

-minor weakness is focus on model system that lacks microbial diversity and may not fully reflect the biology of more complex communities.

Major Comments:

None

Minor Comments:

-As an additional experimental group to assess the role of motility in addition to increased smooth muscle function with ricinoleic acid can you administer an agent that would decrease motility (loperamide) to see if it would augment clustering?

-It would be interesting to probe their model system with addition alterations- increase volume and/or water content with osmotic agents (polyethylene glycol or psyllium husk), or alter nutrient availability with removal of fiber content or additional of purified fiber.

-Fig 3C- the mutants appear to have some fitness cost (lower cell numbers). Does this result in lower total numbers or equal total numbers with more B theta and E rectale?

-Can you include a representative image from the tip as well (either in main or supplemental)?

-The observation that maternal transmission vs post-natal microbial acquisition has dramatic impact on IgA response is very interesting!

-Use of Z1331::cheY lacking chemotaxis and Z1331::flhD lacking flagella was nice

-Line 229 should read: we evaluated the extent of bacterial clustering in the cecum in these three cases.

-I have minor concerns that in mice with a simple gnotobiotic community the enlarged cecum with altered mucus structure may limit the ability to extrapolate these data to SPF mice. This is discussed well in line 400-408 of discussion.

---

## [Decision Letter · Decision Letter 2]

13 Mar 2026

Dear Dr Arnoldini,

Thank you for your patience while we considered your revised manuscript "Emergent spatial structure in the gut microbiota is driven by bacterial growth and gut contractions" for publication as a Research Article at PLOS Biology. This revised version of your manuscript has been evaluated by the PLOS Biology editors, the Academic Editor and the original reviewers.

Based on the reviews, we are likely to accept this manuscript for publication, provided you satisfactorily address the remaining points raised by Reviewer 2. Please also make sure to address the following data and other policy-related requests.

Please supply the numerical values either in the a supplementary file or as a permanent DOI’d deposition for the following figures:

Figure 1CE, 2BD, 3ABDEGH, 4ABCE, S2ABEF, S3ABC, S4A-D, S5AB, S6ABC, S7, S8AB

→ I am aware that your data availability statement provides the link with the data to generate the figures. However, it is not clear which data belongs to which figure. If you use code to generate the figures, could you provide it too? Thank you!

2) Please cite the location of the data clearly in all relevant main and supplementary Figure legends, e.g. “The data underlying this Figure can be found in S1 Data” or “The data underlying this Figure can be found in https://doi.org/10.5281/zenodo.XXXXX”

3) Please ensure that your Data Statement in the submission system accurately describes where your data can be found and is in final format, as it will be published as written there

4) Per journal policy, if you have generated any custom code during the course of this investigation, please make it available without restrictions. Please ensure that the code is sufficiently well documented and reusable, and that your Data Statement in the Editorial Manager submission system accurately describes where your code can be found. More information on our Code Policy, what and how to share can be found here: https://journals.plos.org/plosbiology/s/code-availability

Please note that we cannot accept sole deposition of code in GitHub, or Gitlib, as this could be changed after publication. However, you can archive this version of your publicly available GitHub code to Zenodo. Once you do this, it will generate a DOI number, which you will need to provide in the Data Accessibility Statement (you are welcome to also provide the GitHub access information). See the process for doing this here: https://docs.github.com/en/repositories/archiving-a-github-repository/referencing-and-citing-content

We expect to receive your revised manuscript within two weeks.

*Published Peer Review History*

*Press*

Sincerely,

Melissa

Melissa Vazquez Hernandez, Ph.D.

Associate Editor

PLOS Biology

REVIEWERS' COMMENTS

Reviewer #1:

I thank the authors for making the minor additional revisions I suggested.

Reviewer #2:

The authors have clearly improved the quality of the manuscript by addressing most of the previous comments and implementing the requested changes, when possible.

In the current version, a few minor issues remain concerning figure references. There appears to be a mismatch in the numbering of supplementary figures (there are two figures labeled S3). In addition, several references to subfigures seem incorrect: line 310 should refer to Fig. S6A; lines 314-317 should refer to Fig. 4A; and line 320 should again refer to Fig. S6A.

Overall, I find the results interesting and suitable for publication in PLOS Biology. Some of the conclusions remain somewhat speculative, but this is largely inherent to the nature of the study. Importantly, the manuscript proposes new hypotheses linking bacterial clustering, the rheology of the digesta, and intestinal motility. In my view, the main merit of the paper lies precisely in highlighting these potential connections and proposing conceptual mechanisms that may structure spatial organization in the gut microbiota. By bringing these ideas forward, the article should stimulate further experimental and theoretical work aimed at deciphering the mechanisms involved.

Reviewer #3:

The authors have addressed all of my concerns and questions in this revision.

---

## [Editor Report · Decision Letter 3]

8 Apr 2026

Dear Markus,

Thank you for the submission of your revised Research Article "Emergent spatial structure in the gut microbiota is driven by bacterial growth and gut contractions" for publication in PLOS Biology. On behalf of my colleagues and the Academic Editor, Ken Cadwell, I am pleased to say that we can in principle accept your manuscript for publication, provided you address any remaining formatting and reporting issues. These will be detailed in an email you should receive within 2-3 business days from our colleagues in the journal operations team; no action is required from you until then. Please note that we will not be able to formally accept your manuscript and schedule it for publication until you have completed any requested changes.

PRESS

Sincerely,

Melissa

Melissa Vazquez Hernandez, Ph.D., Ph.D.

Associate Editor

PLOS Biology
